

# Negative emotional state slows down movement speed: behavioral and neural evidence

Xiawen Li[1], Guanghui Zhang[2], Chenglin Zhou[1] and Xiaochun Wang[1]

[1] School of Psychology, Shanghai University of Sport, Shanghai, China
[2] Faculty of Information Technology, University of Jyväskylä, Jyväskylä, Finland

## ABSTRACT

**Background**. Athletic performance is affected by emotional state. Athletes may underperform in competition due to poor emotion regulation. Movement speed plays an important role in many competition events. Flexible control of movement speed is critical for effective athletic performance. Although behavioral evidence showed that negative emotion can influence movement speed, the nature of the relationship remains controversial. Thus, the present study investigated how negative emotion affects movement speed and the neural mechanism underlying the interaction between emotion processing and movement control.

**Methods**. The present study combined electroencephalography (EEG) technology with a cued-action task to investigate the effect of negative emotion on movement speed. In total, 21 undergraduate students were recruited for this study. Participants were asked to perform six consecutive action tasks after viewing an emotional picture. Pictures were presented in two blocks (one negative and one neutral). After the participants completed a set of tasks (neutral of negative), they were subjected to complete a 9-point self-assessment manikin scale. Participants underwent EEG while performing the tasks.

**Results**. At the behavior level, there was a significant main effect of emotional valence on movement speed, with participants exhibiting significantly slower movements in the negative emotional condition than in the neutral condition. EEG data showed increased theta oscillation and larger P1 amplitude in response to negative than to neural images suggesting that more cognitive resources were required to process negative than neutral images. EEG data also showed a larger late CNV area in the neutral condition than in the negative condition, which suggested that there was a significant decrease in brain activation during action tasks in negative emotional condition than in the neural. While the early CNV did not reveal a significant main effect of emotional valence.

**Conclusion**. The present results indicate that a negative emotion can slow movement, which is largely due to negative emotional processing consuming more resources than non-emotional processing and this interference effect mainly occurred in the late movement preparation phase.

Corresponding author
Xiaochun Wang,
wangxiaochun@sus.edu.cn

## INTRODUCTION

Movement speed plays an important role in sports. Flexible control of movement speed is critical for effective athletic performance. Many factors can influence movement speed, with emotional state, especially a negative emotional state, being a potentially important factor. Athletes may underperform in competition due to poor emotion regulation. Although emotion effects on movement speed have been explored previously, the nature of the relationship remains controversial. Elucidation of how negative emotion affects movement speed may be useful toward guiding athletes with respect to how to adjust modulate emotional state according to competition circumstances and regulate their movement speed accordingly.

Studies have demonstrated an intimate relationship between emotion and the motor system (*Beatty et al., 2016*; *Simoes Matos Saraiva, 2017*). Specifically, emotional state has been reported to affect movement speed (*Hälbig et al., 2011*), including walking speed (*Michalak et al., 2009*), running speed (*Lane et al., 2015*), and even the velocity of a thrown ball (*Rathschlag & Memmert, 2013*). The broaden-and-build theory of positive emotions posits that positive emotion can broaden attention and cognitive range, thus promoting agility of thinking (*Fredrickson, 2015*), while negative emotions do the opposite. In accordance with this theory, research has shown that under induced states of sadness and depression, human subjects exhibited an altered gait pattern characterized by reduced walking speeds, vertical head movements, and arm swinging (*Michalak et al., 2009*). In a study employing a target-detection paradigm, *Pereira et al. (2006)* described two types of interference effects of negative emotion on reaction speed: transient and sustained. They found that the duration of interference was related to negative emotional state stability with more stable negative states producing longer lasting interference effects (*Pereira et al., 2006*). It has been suggested that the impact of emotion on movement may be related to the time interval between the emotional stimulus and task-cue stimulus presentation with the effect on movement performance dissipating over time and, potentially, being fully mitigated after a sufficient time interval (*Contreras et al., 2013*).

From an evolutionary point of view, emotional responses may be defensive or appetitive (*Bradley et al., 2001*). The defensive system is activated primarily by negative stimuli triggering avoidance, while the appetitive system is activated by positive stimuli triggering approach. Hence, it is natural for people to approach a favorable stimulus and avoid a harmful one. If movement conforms to this evolutionary rule, speed might become faster (*Chen & Bargh, 1999*). However, unlike other negative emotions, anger has been proved to be an approach-related emotion that can activate the appetitive motivational system (*Carver & Eddie, 2009*). Thus, our experimental materials did not include angry emotional stimuli based on this specificity. The classic paradigm employed in examining this question requires participants to pull or push a joystick after watching a positive or a negative stimulus. Pushing and pulling motions have been found to be faster under negative and positive stimulus conditions, respectively, relative to motions made under neutral conditions (*Krieglmeyer & Deutsch, 2010*; *Rotteveel & Phaf, 2004*). Some studies have demonstrated that emotional stimuli, especially negative stimuli (disgust and fear

stimuli), can activate approach or avoidance actions automatically. However, *Rotteveel & Phaf (2004)* has claimed instead that the influence of emotional stimuli on actions requires conscious awareness. In the present study, participants were required to view the emotional pictures before performing action tasks consciously.

Other prevailing ideas claim that the effect of emotion on movement depends on the specific category of emotion (*Campo et al., in press*; *Rathschlag & Memmert, 2015*). For example, participants threw a ball faster when instructed to recall feelings of anger or happiness than in an emotionally neutral state. Meanwhile, participants jumped higher under anger and happiness conditions than under anxiety and sadness conditions (*Rathschlag & Memmert, 2013*). Taken together, positive emotion has been shown to benefit movement speed control consistently, while the influence of negative emotion on movement speed is more controversial on behavioral level.

It has been shown that some brain circuits responsible for movement control are also involved in emotional processing (*Stoodley & Schmahmann, 2010*). Brain circuits, which were involved in emotional processing, are activated during the movement of athletics (*Lang, Bradley & Cuthbert, 1998*). Viewing emotional pictures has been reported to alter motor cortex excitability (*Coombes et al., 2009*), and negative emotion has been suggested to facilitate primary motor cortex plasticity by modulating intracortical GABAergic neurotransmission (*Koganemaru et al., 2012*). However, electroencephalography (EEG) evidence related to these effects is quite limited. More evidence is needed to understand the neural mechanism underlying the interaction between emotion and movement control.

In the current study, we combined EEG technology with a cued-action task to investigate the effect of negative emotion on movement speed, the duration of this effect, and the ERP correlates of such an interaction. The task used in this study was a combination of the target-detection paradigm developed to investigate how viewing task-irrelevant emotional pictures affects the performance of a detection task (*Pereira et al., 2006*) and a cued-target paradigm, the classic contingent negative variation (CNV) induced paradigm (*Rockstroh et al., 1982*; *Walter et al., 1964*). Unlike the simple button press task of studying response times, the present action task was more complex including three consecutive finger actions and thus more suitable to examine the movement speed. The study analyzed theta oscillation and P1 component in the emotional image processing period and the CNV component in the action task period. CNV has been related to movement preparation for such that CNV amplitude is enhanced when more attention is allocated or a movement is well prepared (*Carretié et al., 2007*; *Walter et al., 1964*). CNV potential consists of early and late CNV arising from the frontal and parietal regions, respectively (*Rohrbaugh, Syndulko & Lindsley, 1976*). The early CNV potential indexes cognitive processing such as sensory preparation, anticipatory attention to the forthcoming target stimuli (*Gómez et al., 2004*; *Loveless & Sanford, 1974*). The late CNV potential corresponds to movement preparation (*Anatürk & Jentzsch, 2015*; *Leuthold, Sommer & Ulrich, 2006*). In addition, occipital theta oscillation and P1 was reported on the presentation of emotion images and researchers observed increased theta and P1 response to negative stimuli (*Aftanas et al., 1998*; *Flaisch et al., 2010*; *Güntekin & Başar, 2014*; *Meng et al., 2016*). We hypothesized that negative emotion would have an interference effect on movement speed. Specifically, we predicted

**Table 1  The SAM Ratings of emotional pictures.**

| Emotion type | Valence $M$ (SD) | Arousal $M$ (SD) |
|---|---|---|
| Negative | 2.63(0.82) | 5.88(1.07) |
| Neutral | 5.03(0.13) | 3.70(1.20) |

that our negative emotion condition would be associated with a slowing of movement speed, theta oscillation and P1 increasing and CNV weakening relative to the neutral condition.

# MATERIALS & METHODS

## Participants

A total of 21 undergraduate volunteers (10 males, 11 females; $21 \pm 2.21$ years) from Shanghai University of Sports participated in this experiment. They were all right-handed, with normal or corrected-to-normal vision, and had no self-reported history of mental illness or chronic physical illness. All of them were paid a modest compensation after the experiment. This experiment was approved by the ethics committee of the Shanghai University of Sport (No. 2015007) and written informed consent was obtained from all participants.

## Materials

We selected 310 emotional images from the Chinese Affective Picture System (*Lu, Hui & Yuxia, 2005*) and International Affective Picture System (*Lang, Bradley & Cuthbert, 1999*), half of which were negative and half neutral. The formal experiment utilized 150 negative and 150 neutral images, and the remaining 10 images were used for practice. Images with the same emotional valence were presented together to induce a stable emotional state (*Larson et al., 2005*). Each picture had been assessed for its valence and arousal on a 9-point self-assessment manikin (SAM) scale. The paired $t$-test performed on the average SAM scores showed that there was a significant difference between negative images and neutral images in emotional valence ($p < 0.001$) and arousal ($p < 0.001$) (Table 1). Negative images had an average pleasure rating of 2.63 and an average arousal rating of 5.88; the neutral images ratings were 5.03 and 3.70, respectively.

## Design and procedure

The experiment employed a $2 \times 2$ within-subject design with emotional valence (negative vs. neutral) and action task sequence (1st and 6th action) as the factors. In the cued-action task, participants were prompted to perform six consecutive action tasks after viewing an emotional picture. Each emotional valence condition consisted of 150 pictures. The task data were compiled by E-Prime 2.0 software.

After being introduced to the cued-action task, participants pressed and held key "2". Simultaneously, an emotional image was presented for 2,000-ms. Participants were instructed to watch this image carefully. The image was followed by an 800-ms presentation of a fixation point. Then, a circle appeared around the fixation point to indicate to the participant that an action was required; once the circle appeared (after 800-ms fixation point), participants were expected to release key "2" immediately, press key "5", and
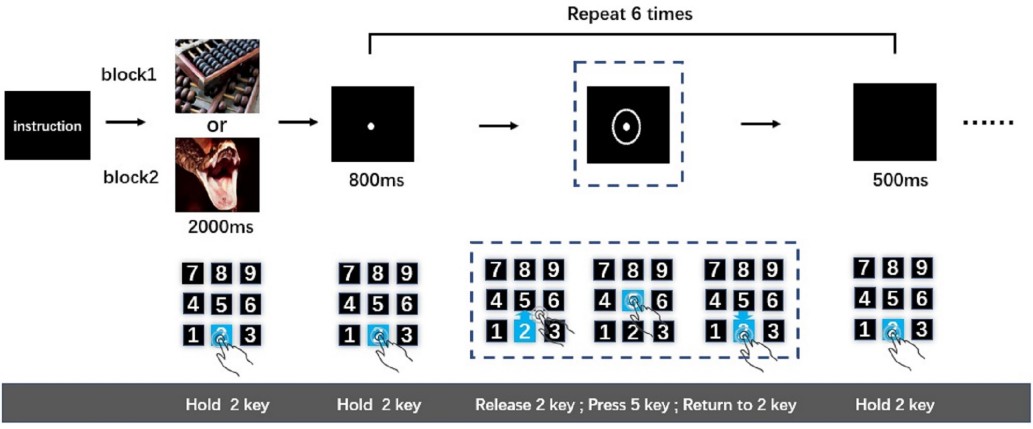

**Figure 1** The cued-action task.

then return to holding key "2". After participants returned to holding key "2", a black screen was presented for 500-ms. This sequence was the first action task. The 800-ms fixation point-action task cue (i.e., circle) 500-ms black screen cycle was repeated five more times resulting in six total action tasks (Fig. 1). This number of action tasks was selected based on previously published work (*Pereira et al., 2006*), wherein it was shown that the analysis of time-separated actions in a series can be used to distinguish sustained from transient interference effects of negative emotion on reaction speed. Only when participants completed the last action task (sixth action task) successfully was the next picture presented. The order of negative and neutral blocks was counterbalanced across participants. After the participants completed a set of tasks (neutral of negative), they were subjected to a 9-point self-assessment manikin scales. The entire experiment was performed in a dimly lit, sound attenuated room.

## Behavior data analyses

Behavioral data of action time were recorded by E-Prime 2.0, including the time spent on the first and the sixth action tasks (the total time it took to release key 2, press 5, release 5, and press 2 again). Data falling beyond three standard deviations of the mean were excluded. Statistical analysis of action time was performed in SPSS 22.0. Data were analyzed by two-way repeated measures analyses of variance (rmANOVAs) with emotional valence (negative and neutral) and action task sequence (1st and 6th action). In order to improve the estimates of the effect, we also conducted a two-way rmANOVAs with emotional valence (negative and neutral) and action task sequence (1st, 2nd, 3rd, 4th, 5th and 6th action).

The scores of the 9-point emotional self-assessment were analyzed by the paired $t$-test with emotional valence (negative and neutral).

## EEG data acquisition and analysis

EEG measures electrical brain responses directly and with high temporal resolution. EEG was conducted with 64 Ag-AgCl electrodes arranged according to the international 10–20

system with a sampling frequency of 1,000 Hz (Brain Products GmbH, Gilching, Germany). The EEG was recorded referentially against the FCz, and AFz served as the ground electrode. The vertical electrooculogram was recorded infra-orbitally at the left eye and the horizontal electrooculogram was recorded latera-orbitally of the right eye. All electrooculogram and electroencephalogram electrodes impedances were maintained below 5 kΩ.

EEG data analysis includes two parts: emotional image processing analysis and action preparation analysis. The EEG were analyzed off-line by EEGLAB in the MATLAB environment. FCz was re-referenced to the average of TP9 and TP10. Then, ocular artifacts were removed through independent component analysis. Next, we remove line noise with a 50 hz notch filter. Then, the data were filtered with a 30-Hz low-pass cutoff and a 0.5 Hz high-pass cutoff, respectively.

Time-domain and time-frequency analysis were used to analyze the emotional image processing data. The data were extracted offline from 200-ms pre-image onset to 2,000-ms post-image onset. All epochs were baseline-corrected with respect to the mean voltage over the 200 ms preceding image onset, epochs with signals that exceeded $\pm 100\,\mu V$ were rejected (95.68% of the trials was retained on average per participant), then averaged by experimental condition. For time-domain analysis, Fast Fourier transform and temporal-Principal Component Analysis (FFT and t-PCA) were performed on data processing (*Achim & Marcantoni, 1997*; *Dien, 2010*; *Dien, 2012*). The data were first filtered with a 0.5∼30 Hz bandpass, and then detection and quantification of the desired ERP components were achieved through a matrix based on t-PCA. P1 at PO3, PO4, PO7,PO8,POz,O1,O2 and Oz within the time window of 190–230-ms was selected as target ERP components (*Luo et al., 2010*; *Müller & Gundlach, 2017*). Averaged P1 amplitude of these electrode was analyzed by the paired $t$-test with emotional valence (negative and neutral). We also selected frontal electrode F3, F4, F5, F6 and Fz to investigated activation differences in different brain regions and conducted a two-way rmANOVAs with emotion valence (negative and neutral) and brain region (frontal and occipital-temporal).

For time-frequency analysis, a complex Morlet continuous wavelet transform (CMCWT) based on the complex wavelet transform (*Tallon-Baudry et al., 1996*) was used for time-frequency analysis of the average ERP data in the MATLAB. CMCWT was described as $CMCWT(t,f) = |(f)*x(t)|^2$. The time-frequency energy $CMCWT(t,f)$ was used to calculate the convolution of the mother wavelet $\Phi(t,fc)$ with the ERP data $x(t)$.

Here, $\Phi(t,fc)$ is the complex Morlet wavelet defined as $\Phi(t,fc) = \frac{1}{\sqrt{\pi\sigma^2}}e^{i2\pi tfc}e^{\frac{-t^2}{2\sigma^2}}$ ($fc$, center frequency; $\sigma$, bandwidth). A wavelet family was characterized by the constant ratio $K = \frac{fc}{\sigma_f} = 2\pi\sigma fc$ with K being greater than 5 (*Zhang et al., 2017*). A baseline correction using the 200-ms preceding image onset again was then conducted. The PO3, PO4, PO7, PO8, POz, O1, O2 and Oz electrode were selected for the analysis of the evoked theta oscillation (5–9 Hz) within the time window of 100–200-ms based on previous study (*Aftanas et al., 2002*). Theta oscillation was analyzed by the paired $t$-test with emotional valence (negative and neutral). We also observed frontal activation at F3, F4, F5, F6 and Fz as well as P1. Two-way rmANOVAs was used to analyze averaged theta oscillations power with emotion valence (negative and neutral) and brain region (frontal and occipital-temporal).

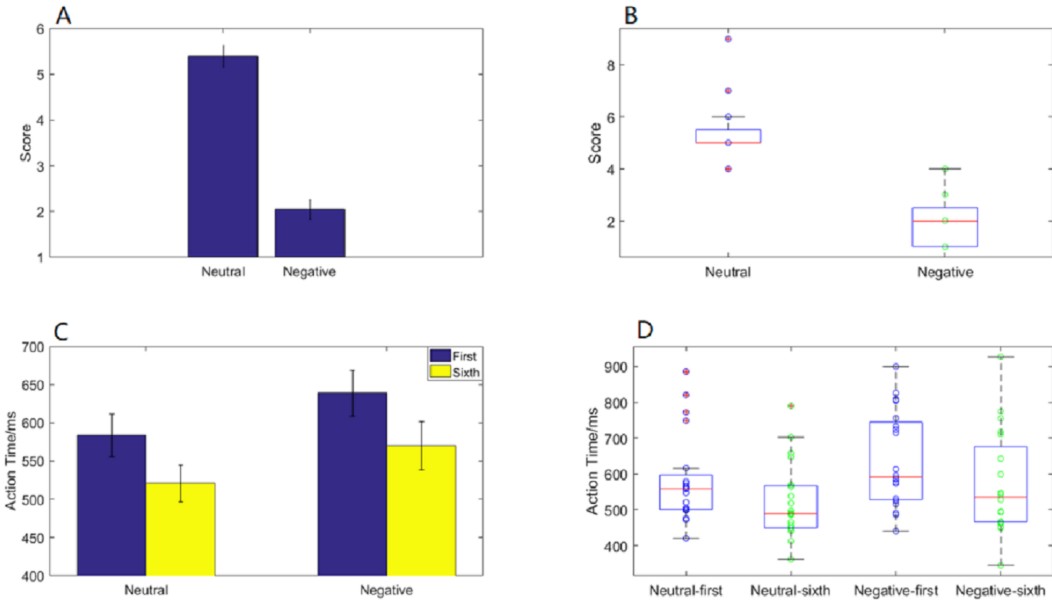

**Figure 2** **Action time "2 × 2 ANOVA" and SAM scores under different experimental conditions.** (A) A histogram of SAM score under neutral and negative conditions. (B) The scatter plots with box plots of SAM score. (C) A histogram of action time under different experimental conditions. (D) The scatter plots with box plots of action time.

For action preparation analysis, the data were segmented from 800-ms prior to the onset of the action task cue (circle) to 1,000-ms after stimulus onset. All epochs were baseline-corrected with respect to the mean voltage over the −800∼−700-ms period preceding stimulus onset, then averaged by experimental condition. CNVs were selected as target ERP components. We analyzed the average early CNV area within the time window of −200-ms∼0-ms at Fz and at the late CNV area within the time window of 0∼140-ms at FCz (*Rohrbaugh, Syndulko & Lindsley, 1976*). After data preprocessing, data of one participant was removed due to many additional movements. We then averaged the participants' data for each period. The average of early CNV and the averaged late CNV data were analyzed by two-way rmANOVAs with emotional valence (negative and neutral) and action task sequence (1st and 6th action) based on the behavioral results.

## RESULTS

### Behaviors

Emotional self-assessment data results showed that the SAM scores in negative condition were significantly less than neutral condition ($p < 0.001$) (Fig. 2A). We used scatter plots with box plots to improve data transparency in Fig. 2B.

Behavioral data results of 2×2 rmANOVAs showed a main effect of both action sequence ($F_{1,19} = 49.807$; $p < 0.001$; $\eta_P^2 = 0.724$) and emotional valence ($F_{1,19} = 6.826$; $p = 0.017$; $\eta_P^2 = 0.264$) on action time. The duration of the sixth action was significantly shorter than that of the first, and action time in the negative condition was significantly longer than

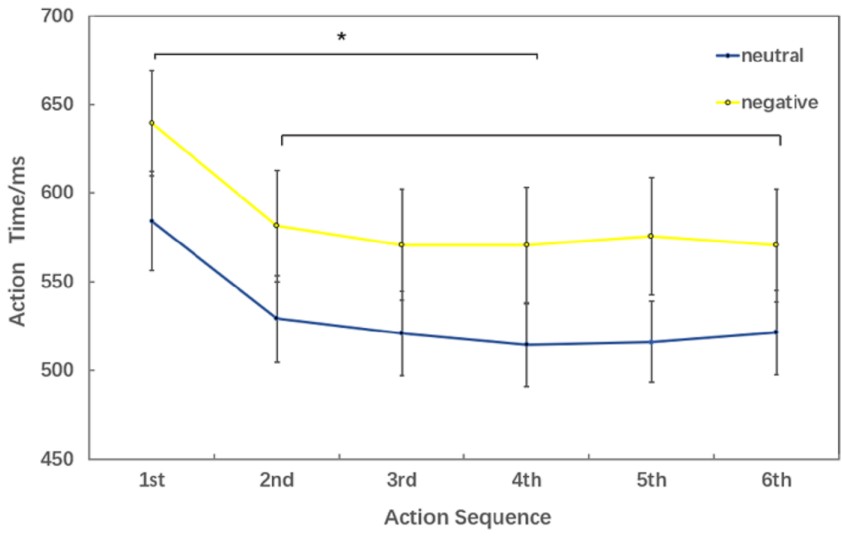

**Figure 3** Action time under different experimental conditions (six levels in the action sequence factor).

that in the neutral condition. However, there was not a significant interaction of these two factors ($F_{1,19} = 0.329$; $p = 0.573$; $\eta_P^2 = 0.017$) (Fig. 2C). The scatter plots with box plots were showed in Fig. 2D.

Behavioral data results of $2 \times 6$ rmANOVAs showed a main effect of both action sequence ($F_{5,15} = 17.467$; $p < 0.001$; $\eta_P^2 = 0.853$) and emotional valence ($F_{1,19} = 7.205$; $p = 0.015$; $\eta_P^2 = 0.275$) on action time. There was not a significant interaction of these two factors ($F_{5,5} = 2.804$; $p = 0.055$; $\eta_P^2 = 0.483$). The pairwise comparison found that the first action time was significantly slower than next 5 action task(2nd, 3rd, 4th, 5th, 6th) ($p < 0.01$); while there was no significant difference between the sixth action task and the previous four action tasks (2nd, 3rd, 4th, 5th) (Fig. 3).

## EEG
### P1 evoked by emotional image
P1 brain topography illustrated that this effect was particularly robust in the occipital-temporal cortex. Paired $t$-test result showed that negative stimuli evoking larger P1 amplitude than neutral stimuli in the occipital-temporal ($p = 0.002$) (Fig. 4A). $2 \times 2$ rmANOVAs of emotion and brain region showed a main effect of brain region ($F_{1,19} = 135.202$; $p < 0.001$; $\eta_P^2 = 0.877$), but not emotion ($F_{1,19} = 3.449$; $p = 0.079$; $\eta_P^2 = 0.154$). P1 evoked in occipital-temporal was significantly larger than that evoked in frontal region. There was a significant interaction between these two factors ($F_{1,19} = 8.446$; $p = 0.009$; $\eta_P^2 = 0.308$) (Figs. 4B and 4C).

## Theta oscillation evoked by emotional image
Brain topography of the evoked theta oscillation (5–9 Hz) was shown in Figs. 5C and 5D. Overt theta oscillations were present at 100–200-ms in negative and neutral condition in occipital-temporal region. Paired $t$-test result showed that negative stimuli evoking larger

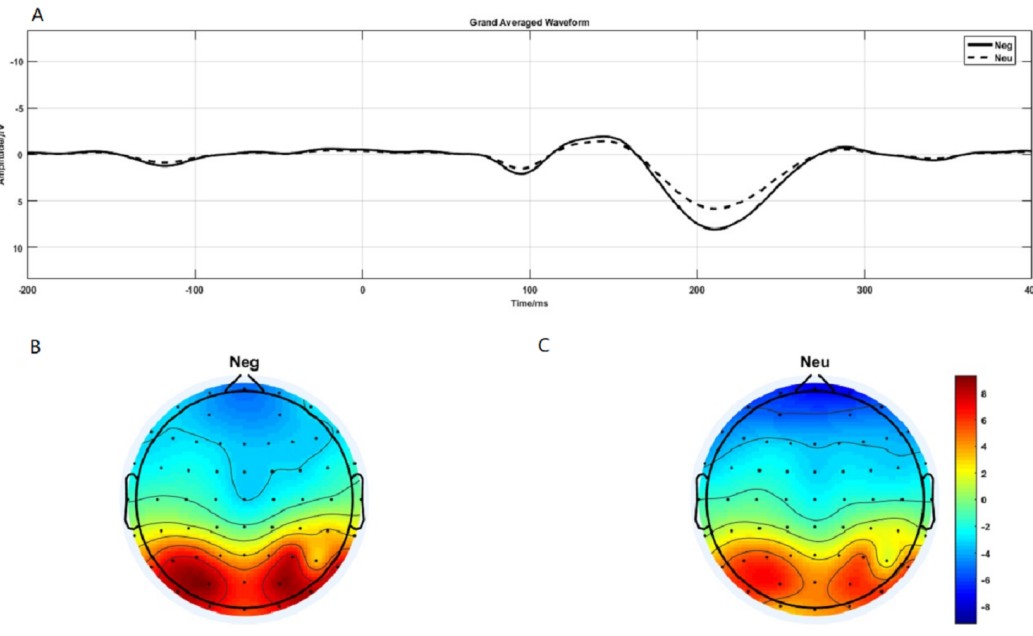

**Figure 4** **P1 component elicited under neutral and negative conditions at electrode sites PO3, PO4, PO7, PO8, POz, O1, O2, Oz and Brain topography of P1.** (A)The P1 component elicited by neutral and negative conditions. (B) Brain topography of P1 under negative condition. (C) Brain topography of P1 under neutral condition.

theta oscillation than that of neutral stimuli ($p = 0.006$) (Figs. 5A and 5B). $2 \times 2$ rmANOVAs of emotion and brain region revealed a main effect of brain region ($F_{1,19} = 9.402$; $p = 0.006$; $\eta_{P2}^2 = 0.331$) and emotion ($F_{1,19} = 10.875$; $p = 0.004$; $\eta_P^2 = 0.364$). Theta oscillation evoked in occipital-temporal was significantly larger than that evoked in the frontal region (Figs. 5C and 5D). Negative stimuli evoked larger theta oscillation than that of neutral stimuli. There was not a significant interaction between these two factors ($F_{1,19} = 1.725$; $p = 0.205$; $\eta_P^2 = 0.083$).

## Early CNV

ERP results revealed a main effect of action sequence ($F_{1,19} = 7.084$; $p = 0.015$; $\eta_P^2 = 0.272$), but not emotion ($F_{1,19} = 2.277$; $p = 0.148$; $\eta_P^2 = 0.107$), on early CNV, with the sixth action condition evoking a larger early CNV area than the first action (Fig. 6A). There was not a significant interaction between these two factors ($F_{1,19} = 0.874$; $p = 0.361$; $\eta_P^2 = 0.044$). Early CNV brain topography illustrated that this effect was particularly robust in the frontoparietal region (Figs. 6C, 6D, 6E, and 6F).

## Late CNV

ERP results revealed a main effect of both emotion ($F_{1,19} = 5.644$; $p = 0.028$; $\eta_P^2 = 0.229$) and action sequence ($F_{1,19} = 22.891$; $p < 0.001$; $\eta_P^2 = 0.546$), on the late CNV, with a larger CNV area in neutral condition than in negative condition and a larger CNV area in sixth action condition than in first action condition (Fig. 6B). There was not a significant interaction between these two factors ($F_{1,19} = 0.347$; $p = 0.564$; $\eta_P^2 = 0.018$). Late CNV

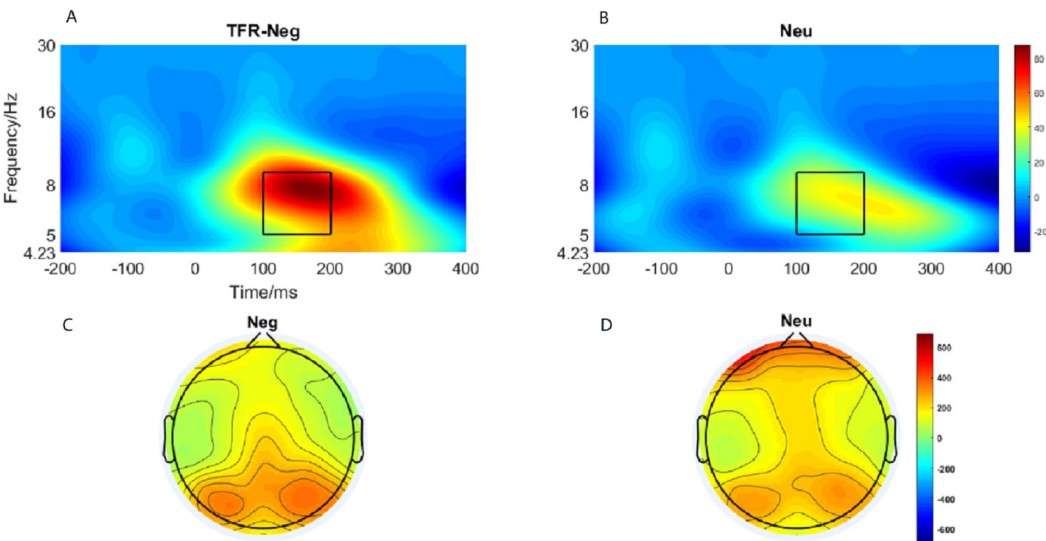

**Figure 5 Theta oscillationelicited under neutral and negative conditions at electrode sites PO3, PO4, PO7, PO8 , POz, O1, O2, Oz andBrain topography of Theta oscillation.** (A) Theta oscillation elicited by negative condition. (B) Theta oscillation elicited by neutral condition. (C) Brain topography of theta oscillation under negative condition. (D) Brain topography of theta oscillation under neutral condition.

brain topography illustrated that this effect was also particularly robust in the frontoparietal region (Figs. 6G, 6H, 6I, and 6J).

# DISCUSSION

The present results support our hypothesis that negative emotion can have an interference effect on movement speed. Our study demonstrates that viewing negative emotion images can induce negative emotion which significantly slowed movement speed compared with that under neutral condition. These behavioral effects were accompanied by significant changes in ERP component. Negative images evoked larger P1 and theta oscillation than neutral images during emotion processing period. Late CNV differed significantly between emotion conditions and action sequence conditions in the action task period. Interestingly, early CNV which is related to cognitive processing was only differed significantly between action sequences rather than emotion conditions.

On the behavioral level, it was shown that viewing negative emotion images clearly induce negative emotion, consistent with the prior finding showing that the magnitude of negative emotion produced by negative images increases as participants view more emotionally negative images (*Pereira et al., 2006*). As a consequence, movement speed in the negative emotion was slowed, compared with that in neutral condition. Our findings are consistent with previous researches claiming that negative emotion may interfere with behavioral performance (*Gross, Crane & Fredrickson, 2012*; *Van Lieshout et al., 2014*). In the negative emotional condition, movement speed was slowed down in the both first action and sixth action task. This means the duration of this slow down effect can be maintained until the end of six action tasks. This interference effect is probably caused by the induced

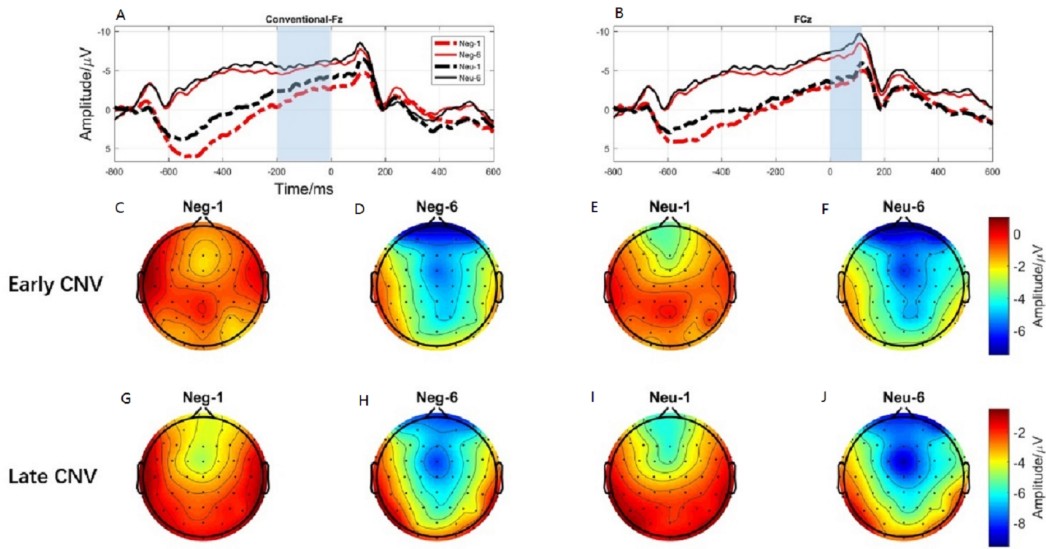

**Figure 6** **The early CNV and late CNV elicited under different conditions at electrode sites Fz, FCz and Brain topography of early CNV and late CNV, respectively.** (A) The early C NV under different conditions at Fz. (B) The early C NV under different conditions at FCz. (C) Brain topography of early CNV at negative emotion and 1st action task. (D) Brain topography of early CNV at negative emotion and 6th action task. (E) Brain topography of early CNV at neutral emotion and 1st action task. (F) Brain topography of early CNV at neutral emotion and 6th action task. (G) Brain topography of late CNV at negative emotion and 1st action task. (H) Brain topography of late CNV at negative emotion and 6th action task. (I) Brain topography of late CNV at neutral emotion and 1st action task. (J) Brain topography of late CNV at neutral emotion and 6th action task.

negative emotion. *Fredrickson (2015)* has shown that individuals in a negative emotional state have a narrower range of cognitive resources, which in turn has a negative impact on individual behavior. *Pereira et al. (2006)* reported both transient and sustained interference effects of negative emotion on a target-detection task. The interference on movement speed can be explained as a reduced availability of cognitive resources (*Fredrickson, 2015*; *Melcher et al., 2012*; *Van der Stigchel, Imants & Ridderinkhof, 2011*), thus disturbing the action task. People are very sensitive to negative stimuli (*Lu et al., 2015*; *Yuan et al., 2007*), with negative stimuli being more likely to attract people's attention and thus more likely to be encoded well (*Zhang et al., 2014*). Attentional blink is a phenomenon wherein people tend to devote more attention to the first stimulus than the second when two stimuli are presented in series. However, *Anderson & Phelps (2001)* observed that if the second stimulus was negative, people were less likely to forget it (*Anderson & Phelps, 2001*), consistent with the notion that negative stimuli tend to attract attention.

In addition to the explanation that negative emotion reduced the availability of cognitive resources, the interference effect on the first action task can be explained from a biological evolutionary perspective. Human behavior is motivated by defensive and appetitive systems (*Bradley et al., 2001*); the defensive system is activated primarily by negative stimuli triggering avoidance, while the appetitive system is activated by positive stimuli triggering approach. It is natural for people to approach a favorable stimulus and avoid

a harmful stimulus. Once people are instructed to approach a negative stimulus, the movement speed will slow down (*Chen & Bargh, 1999*; *Rotteveel & Phaf, 2004*). Our action task can be viewed as an approach movement to the images (*Maxwell & Davidson, 2010*). 2004). If participants are going to approach a negative image, the action would be more disturbed than a neutral image.

Behavioral results also found that movement speed was significantly slower in the first action task than in the sixth action task. This effect may be partly due to images presented before the first action (interference of the first action) and partly due to the practice effect (promotion of the six action). Participants were easily distracted by task-irrelevant stimuli (images), which interferes with the processing of the upcoming action-cued target (*Contreras et al., 2013*), and thus affects movement speed. Thus, less attention resources were devoted to the first action task than the sixth action task. *Posner & Petersen (2012)* argued that in order to orient to a new task, individuals first had to disengage from what they were currently focusing on (*Posner & Petersen, 2012*). The time interval between the emotion stimuli and action-cued stimuli (800-ms) in our experiment was within the purported attentional range period (500–800-ms) (*Müller, Tedersälejärvi & Hillyard, 1998*). The presumption is that such interference effects are due to participants' attention being partially consumed by the emotional images preceding the first action task. Our present observation of an interference effect in a relatively long time interval of 800-ms suggests that the attentional range may persist for a period of at least 800-ms. In addition, the repetition of action tasks will lead to a practice effect which may also improve the movement speed. Thus, both the interference effect of images before the first action task and the practice effect on the sixth action task will make participants to perform the sixth action task faster than perform the first action task. ERP data will further reveal the brain processing mechanisms underlying this behavior.

The behavioral data of $2 \times 6$ rmANOVAs showed a more comprehensive picture of how negative emotion affected movement speed. In all six action tasks, movement speed in negative conditions was significantly slower than that in neutral conditions which is in consistent with that in $2 \times 2$ rmANOVAs. According to the comparison between different action sequences (there was no significant difference between the last four action time), we speculated that the sixth action had reached a steady state and the brain processing mechanism was similar between the last four action tasks. That's why we chose $2 \times 2$ rmANOVAs in the movement related ERP analysis

At the neural level, we observed distinct time-domain P1 component in the time window of 190-230-ms and theta oscillations(5–9 Hz) in the time window of 100-200-ms during the emotion processing period. The present brain topography for P1 and theta oscillations results suggest that the activating region was located in the occipital-temporal cortex, which is responsible for visual processing (*Wong et al., 2009*). The results showed that the P1 and theta oscillations elicited in the negative condition was significantly larger than that in the neutral condition. P1 was a occipital-temporal positive component elicited by the onset of emotional faces and scenes images (*Luo et al., 2010*; *Müller & Gundlach, 2017*). Relative to neutral images, negative images elicited a larger positive P1 amplitude (*Hammerschmidt et al., 2018*; *Luo et al., 2010*). *Flaisch et al. (2010)* also found that negative gestures elicited

increased P1 than neutral gestures (*Flaisch et al., 2010*). Researchers also found that P1 was regulated by attention resources (*Taylor, 2002*). Thus, the present larger P1 component elicited by negative stimuli suggesting that negative image processing consume more attention resources.

Past studies have shown that evoked theta oscillations in the occipital-temporal cortex was sensitive to emotional valence and arousal (*Aftanas et al., 2002*). Elevated parietal-occipital theta responses were found on presentation of negative stimuli (*Balconi, Brambilla & Falbo, 2009*). *Sun et al. (2012)* also found that negative stimuli elicit higher theta responses than neutral cues (*Sun et al., 2012*). However, researchers also reported increased theta response on cognitive load in the frontal area (*Güntekin & Başar, 2009*; *González-Roldan et al., 2011*). Hence, considered in the context of the present study, a larger P1 and theta oscillations elicited in the negative condition suggest that more cognitive resources are allocated to negative image processing than to neutral image processing, reducing the availability of attention resources for the action tasks, leading to a reduction of movement speed.

Consistent with the behavioral finding that movement speed was significantly faster in the sixth action task than in the first, ERP data revealed a larger CNV component (early CNV and late CNV), in the sixth action task than that in the first. Past studies have shown that CNV was related to anticipatory, attention distribution and movement preparation (*Carretié et al., 2007*; *Walter et al., 1964*). The CNV results suggested that participants paid more attention to the sixth action task than that in the first action task. In the present experiment, images were presented before the first action cue, while the sixth action was preceded by a blank screen. The presence of images was a disturbance to the first action task. Participants might not be able to disengage their attention from images to the first action cue. Previous studies have shown a response slowing (*Anatürk & Jentzsch, 2015*) and an increase in CNV (*Arjona, Escudero & Gómez, 2014*) in trials preceded by valid cues compared to invalid cues, consistent with our results. *Lin et al. (2014)* provides evidence that uncertain cues produced smaller early contingent negative variation (CNV) than did the certain cues about upcoming task. Their CNV results are also in accordance with their behavioral findings that participants performed better under certain cues than uncertain cues (*Lin et al., 2014*). Furthermore, the preparation phase for the sixth action started from -600-ms before the target onset, 400-ms earlier than the first action (−200-ms) (Fig. 3). This finding proved that the sixth action was well prepared than the first action.

Brain topography for CNVs showed that CNV activating region was centered over the parieto-frontal cortex, an area that is important for attentional processes as well as motor planning, preparation, and execution. The CNV component has been shown to be closely linked with attention (*Faugeras & Naccache, 2016*; *Tecce, 1972*), motor preparation (*Carretié et al., 2007*). Participants were in the motor preparation phase throughout the time period preceding presentation of the task cue (circle). The time window of the CNV corresponded exactly to this preparation period. Previous studies have shown that the larger the amplitude elicited during movement preparation, the faster the resulting action was (*Anatürk & Jentzsch, 2015*; *Walter et al., 1964*), consistent with our results. Therefore, our

significant CNV component findings may reflect the participation of CNV in movement preparation and attention distribution.

Interestingly, the main effect of emotion condition was only observed in the late CNV component, with neutral condition elicited larger late CNV amplitude than that elicited in the negative condition. Past studies have shown that late CNV evoked in the parietal cortex was closely related to movement preparation (*Gaillard, 1985*). Enough attention resources must be involved in order to complete the movement with high quality (*Shaw et al., 2018*). CNV has been proven to be positively related to the attention allocation. Studies have shown that individuals with attention deficit evoked smaller CNV amplitude than that evoked by normal people in attention related tasks (*AlbrechtDaniel et al., 2014*). Researchers also found that attention training, like mindfulness, could increase the activation of CNV (*Bostanov et al., 2018*). Thus, CNV is always considered as a measure of increased concentration. However, evidence showed that attention resource allocation to negative stimuli was significantly different with neutral contents (*Buodo, Sarlo & Palomba, 2002*). *Vanlessen et al. (2015)* using an anti-saccade task revealed that neutral condition evoked larger CNV component than affective condition (*Vanlessen et al., 2015*). *Cudo et al. (2018)* also demonstrated that CNV amplitude was less negative in the high than in the low-approach motivated affect (*Cudo et al., 2018*). Hence, the present late CNV effect suggests that more cognitive resources were required when processing negative emotion, disturbing the motor preparation phase of the simultaneous action task.

The brain topography results of late CNV suggest that the activating region was located in the parietal cortex, which is responsible for movement processing (*Wong et al., 2009*). Considered in the context of the present study, a smaller late CNV elicited in the negative condition suggests that more cognitive resources are allocated to negative emotion processing, than to neutral processing, reducing the availability of attention resources for the action tasks, leading to a reduction of movement speed.

The present study had a few limitations. Firstly, the total number of participants was relatively small. The results might be influenced by individual differences. Secondly, in order to ensure an induced negative emotion, participants were subject to viewing 150 images. Each image was followed by six action tasks, which may lead to a fatigue effect. Thirdly, participants were subjected to press key 2 all the time while viewing images as mentioned above. This press action particularly disturbed the EEG signal extraction. In addition, the present study did not observe interaction of emotional valence and action sequence. We can increase the number of action tasks (like 12 action tasks) after each emotional image in future research.

## CONCLUSIONS

Our study demonstrates that movement speed was slowed in the negative emotional state. In addition, our ERP results support that the cognitive resources consumed by negative emotion processing may diminish the availability of attentional resources for action tasks. Taken together, the present behavioral and EEG data in the present study suggest that negative emotion has an interference effect on movement speed.

### Funding

This study was supported by a grant from Project 31500911 funded by the National Natural Science Foundation of China. The funders had no role in study design, data collection and analysis, decision to publish, or preparation of the manuscript.

### Grant Disclosures

The following grant information was disclosed by the authors:
National Natural Science Foundation of China: 31500911.

### Competing Interests

The authors declare there are no competing interests.

### Author Contributions

- Xiawen Li conceived and designed the experiments, performed the experiments, analyzed the data, prepared figures and/or tables, authored or reviewed drafts of the paper, approved the final draft.
- Guanghui Zhang analyzed the data, prepared figures and/or tables.
- Chenglin Zhou and Xiaochun Wang conceived and designed the experiments, contributed reagents/materials/analysis tools.

### Human Ethics

The following information was supplied relating to ethical approvals (i.e., approving body and any reference numbers):

This study received approval from the Ethics Committee of Shanghai University of Sport (2015007).

### Ethics

The following information was supplied relating to ethical approvals (i.e., approving body and any reference numbers):

This study received approval from the Ethics Committee of Shanghai University of Sport (2015007).

### Data Availability

The raw data are available in the Supplemental Files.

### Supplemental Information

Supplemental information for this article can be found online at http://dx.doi.org/10.7717/peerj.7591#supplemental-information.

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
