# Peer review of "Negative emotional state slows down movement speed: behavioral and neural evidence"

_PeerJ, doi:10.7717/peerj.7591_

## Round 0.1 · original submission · Major Revisions

The reviewers provided constructive feedback that the authors should use to further improve their manuscript. In particular, the authors should plot data from individual participants, report the statistical results in full, justify the selection of EEG channels and reconsider the statistical tests that are used. It would also be helpful to include a list of the stimuli that were used and relabel the SPSS files to English.

·

Basic reporting

There are some minor typos and language issues throughout the document. For example:

Abstract-Conclusion – “emotional processing consuming more resources” should specify cognitive resources.
Line 58 – “Neural condition” was used instead of “neutral condition”.
Line 152 – should say “researchers”
Line 324 – “For the sixth action task, the interference effect may attribute to the induced negative emotion.”
Line 416 – “Study have showed that attention-deficit individuals evoked lower CNV amplitude than normal.”
Line 435 – “Thus, the total number of action tasks was a little much. there might be a fatigue effect.”
Line 439 – “Maybe we can increase the number of action tasks (like 12 action task) after each image viewing and reduce total emotional images.”

I’m unsure of the relevance of EPN and LPP (Line 367), as they weren’t discussed in the Introduction, or mentioned in the Methods as EEG components of interest. Did the authors also examine EPN and LPP? If so, this should be reported, and there should be an appropriate statistical correction for multiple comparisons.

I encourage the authors to consider plotting the data in Figure 2-3 such that the raw scores can be observed for all time points e.g. by using scatter plots with box plots. Efforts made to improve data transparency will certainly aid future interpretations of the findings. An exploration of the benefits of transitioning away from bar graphs can be read here:
https://doi.org/10.1371/journal.pbio.1002128
and
https://garstats.wordpress.com/2016/03/09/one-simple-step-to-improve-statistical-inferences/

Experimental design

The article is written to address the question of movement speed, particularly with regards to sport. However, the experiment design is better suited to examine response time, which is also a very important factor for athletes. In the present study, participants are required to perform a relatively simple button press task in which the limiting factor is not likely to be finger movement speed, but rather reaction time to visual stimuli. Perhaps this distinction needs to be made clearer in the Introduction.

To aid researchers who wish to replicate this research, the exact image file names used from the CAPS and IAPS databases should be provided in Supplementary Materials.

Validity of the findings

Line 170 – It appears as though the images extracted from CAPS were mixed with those from the IAPS. Did the authors conduct statistical testing to ensure that the valence and arousal ratings of the images from both databases were approximately equivalent? Additionally, it would be useful to know the standard deviations for valence and arousal ratings of the negative and neutral images.

Regarding EEG analyses, were EEG epochs checked for noise artefacts? If so, how many trials were removed on average per participant? Did the authors perform an independent component analysis (ICA) to remove muscle and blink artefacts?

The statistical analysis makes use of a 2 x 2 rmANOVA. However, it seems to me that a general linear model with a categorical factor of emotional valence (negative vs neutral) and a continuous factor of task sequence (1st through to 6th) would be more appropriate. This has added benefit of incorporating data from the 2nd, 3rd, 4th and 5th cued-action tasks, thereby improving estimates of the effect.

Line 316 – The difference between 1st and 6th action tasks is interpreted in the manuscript as competition for cognitive resources during the 1st task, which dissipates by the 6th task. However, it is possible that at least part of the increase in speed from 1st to 6th task is a practice effect. This could be checked by examining the discrepancy in response time between 1st and 6th trials for the first 75 images displayed as compared to the last half of the images. A practice effect would show greater discrepancy 1st - 6th actions within the first half of images compared to the last half due to ceiling effects. A distractor effect would reveal relatively consistent discrepancies throughout the trials. A general linear model using continuous data for the action sequence tasks could be used to quantify the strength of distractor and practice effects using the following fixed factors:
• Emotional Valence (negative vs neutral)
• Action sequence (1st, 2nd, 3rd, 4th, 5th, and 6th)
• Order (continuous labelling of images in the order displayed. Can also be turned into a categorical variable by labelling images as halves/tertiles/quartiles)
An analysis of this type would provide additional support for the discussion in Line 324 e.g. “for the sixth action task, the interference effect may attribute to the induced negative emotion”.

Line 235 – It’s not clear to me why a one-way rmANOVA was used to analyse averaged P1 amplitudes. If the data is grouped using one factor (emotion valence) with two levels (negative and neutral), a paired t-test would be appropriate. The same applies to averaged theta oscillations (line 247).

Additional comments

Line 338 – is it not also possible that negative stimuli reduce the availability of cognitive resources for the 1st action task as well as the 6th, thus consistently slowing response time? This appears to be the most parsimonious explanation that doesn’t require invoking pre-activation of motor systems through negative imagery. More exposition is needed to make the reader understand why this simple explanation doesn’t fit the observed data.

Line 347 – “Behavioral findings that movement speed was significantly slower in the first action task than in the sixth action task was also remarkable.” I would tone down the language here (i.e. the word ‘remarkable’). Especially as this result is entirely in line with expectations – participants show improved response times with repetition on the action tasks due to practice effects. This is also supported by the CNV results, which showed improved attention and preparation in later action tasks relative to earlier tasks.

Line 370 – “This might be due to the interference of the action task.” I’m not sure what this sentence refers to, it does not seem to relate directly to the preceding sentence.

·

Basic reporting

Title: Negative emotional state slows down movement speed: behavioral and neural evidence

The manuscript by Li and Colleagues analyzed the effect of emotions on movement speed. The authors presented emotional pictures to the subjects during EEG recording and evaluated the speed of these subjects after each picture. The authors reported that EEG data showed increased theta oscillation and larger P1 amplitude in response to negative than to neural images. Furthermore, participants exhibited significantly slower movements in the negative emotional condition than in the neutral condition.

In general, the manuscript has intriguing results and merits publication. However, there are also some points that should be clarified before publication.

The introduction is well written I do not have any critics for the introduction part.

Experimental design

Materials line 176: The results of SAM should be given in the results section with more information, including the means, Standard deviations, and comparisons between negative and neutral emotions with p values.

Line 224: FCz was re-referenced to the average of TP9 and TP10. It is not clear why the authors re reference the data to these specific electrodes?

Line 233: “P1 at PO3, PO4, PO7 and PO8 within the time window of 190-230-ms was selected as target ERP” and line 245 “The POz,PO7,and PO8 electrode were selected for the analysis of the evoked theta oscillation (5–9Hz) within the time window of 100- 200-ms”. This point is the weakest point of all paper. The authors just select few electrodes to analyse and these electrodes are not same for the analysis of P1 and theta. Although authors give reference to Aftanas work as a rationale it is still not understood. There are many other papers which analyze the event related oscillations in the whole topology. There could be hemispheric differences or dominance of occipital location in comparison to frontal region. Although the theta responses are the dominant rhythm of frontal locations during emotional pictures theta responses also increase in parietal and occipital locations. The authors only analyze the parietal and parietal-occipital and they even do not analyze the occipital electrodes (O1, Oz, O2). This point is an important limitation of the study.

Line 360 Although the authors only analyze parietal electrodes they discuss the importance of occipital electrodes. The analysis should also include occipital electrodes the authors then may discuss this point.

Validity of the findings

It will be nice to see the results of correlation analysis between the SAM scores and theta responses and as well as the speed of movement.

The additional SPSS files presented as a data file does not help to the reviewers like me who cannot read and talk Chinese.

---

## Round 0.2 · Minor Revisions

The reviewer pointed at a few typos that need to be corrected before the manuscript can be accepted for publication.

·

Basic reporting

There remain a few minor typos in the document:

Line 109: speed might become faster than violate this rule
Line 130: Brain circuits involved in emotional processing are activated during the movement of athletics by trained athletes.
Line 178: …negative images were significantly differed from neutral images.
Line 168: After data preprocessing, a subject with many additional movements were removed.
Line 277: We used [the] scatter plots with box plots to improve data transparency in Fig.2B.
Line 344: Our findings are in line with previous study claiming.
Line 346: The significantly decrease of negative emotion.
Line 363: …the interference effect for the first action task can be explained by biological evolutionary perspective.
Line 368: Once people were instructed to approach a negative stimulus, the movement speed will be slowed down.
Line 387: …the practice effect on the sixth action task will make participants to perform the sixth action task faster.
Line 390: …behavior data results of 2×6 rmANOVAs add us understand more comprehensively.
Line 425: …greater dedication of attentional resources and well movement preparation
Line 449: …with neutral condition elicited larger CNV than negative condition.
Line 453: Study have showed that attention-deficit individuals.

The use of the word “besides” to begin a sentence is a touch too colloquial/informal. It can also be misinterpreted to mean either “additionally…” or “but more to the point…”, where the former is the continuation of a thought, and the latter is a partial negation.

The legends in Figure 3, 4, and 5 says “neural” instead of “neutral”

Experimental design

I very much appreciate that the authors re-analysed the data using a 2 x 6 rmANOVA in addition to the original 2 x 2 rmANOVA.

Validity of the findings

The Discussion is well thought through, and the findings appear to be well-supported by the data.

·

Basic reporting

Authors have addressed my remarks adequately, and I have no other comments. The manuscript merits publication in its revised form.

Experimental design

Authors have addressed my remarks adequately, and I have no other comments.

Validity of the findings

Authors have addressed my remarks adequately, and I have no other comments.

Additional comments

Authors have addressed my remarks adequately, and I have no other comments.

---

## Round 0.3 · accepted · Accept

The authors have adequately addressed the outstanding comments.